# A New Clue for the Late Eocene Freshwater Ecosystem of Central China Evidenced by New Fossils of *Trapa* L. and *Hemitrapa* Miki (Lythraceae)

**DOI:** 10.3390/biology11101442

**Published:** 2022-10-01

**Authors:** Zhuochen Han, Hui Jia, Xiangning Meng, David K. Ferguson, Mingyue Luo, Ping Liu, Junjie Wang, Cheng Quan

**Affiliations:** 1School of Earth Sciences and Engineering Xi’an Shiyou University, Xi’an 710065, China; 2Shaanxi Key Lab of Petroleum Accumulation Geology, Xi’an Shiyou University, Xi’an 710065, China; 3State Key Laboratory of Palaeobiology and Stratigraphy, Nanjing Institute of Geology and Palaeontology, Nanjing 210008, China; 4Department of Paleontology, University of Vienna, 1010 Vienna, Austria; 5School of Earth Science and Resources Chang’an University, Xi’an 710054, China

**Keywords:** *Trapa*, *Hemitrapa*, paleoenvironment, late Eocene, central China

## Abstract

**Simple Summary:**

This paper describes the aquatic plants of *Trapa* L. and *Hemi**trapa* Miki from the upper Eocene of Bailuyuan Formation in the Weihe Basin, central China. The discoveries represent the earliest known *Trapa* records in the world and the earliest *Hemitrapa* record in Asia. Furthermore, the new species identified in this study is the most reliable leaf fossil record of *Trapa* so far. These occurrences provide a new clue to investigate the origin of *Trapa* and its evolutionary relationships with *Hemitrapa*. The unexpected aquatic plant assemblage indicates that central China was warm and humid, with freshwater ponds or lakes, in the late Eocene.

**Abstract:**

Both *Trapa* L. and the extinct *Hemitrapa* Miki are aquatic plants in the family Lythraceae, with abundant fossil records in Eurasia and North America in the Cenozoic. However, documented materials are mainly based on fruit and pollen grains without reliable leaf fossils. Here, we report fossil leaves, fruit, and roots of *Trapa* and fruit of *Hemitrapa* from the late Eocene of Weinan, the Weihe Basin of central China. The fossil leaves are identified as a new species, *Trapa natanifolia* Z. C. Han et H. Jia sp. nov., which represents the earliest known record of a *Trapa* leaf fossil. It is remarkably similar to extant species of *Trapa*, mostly due to the unique inflated petiole structures found in both of them. While displaying prominent intergeneric differences, the incomplete fossil fruits are assigned to *Trapa* sp. indet. and *Hemitrapa* sp. indet. The former is the earliest fossil fruit record of *Trapa*, and the latter represents the earliest fossil record of *Hemitrapa* found in Asia. These new fossil discoveries suggest that the divergence of *Trapa* and *Hemitrapa* occurred at least by the late Eocene. It is believed that modern *Trapa* most likely originated in China. Furthermore, this unexpected aquatic plant fossil assemblage indicates that central China was warm and humid, with freshwater ponds or lakes, in the late Eocene and not as arid as previously thought.

## 1. Introduction

*Trapa* L. is a floating annual herb that grows in slow-motion freshwater bodies such as lakes, ponds, and slow rivers. It is widely distributed from the tropical to warm temperate regions of Eurasia and Africa [1]. *Trapa* is considered to be strongly related to the family Lythraceae based on the results of a multigene phylogenetic analysis [2]. The generic characteristics of *Trapa* are remarkable and easily recognized. However, the morphological diversity in the genus and the large variation in structural traits have led to confusion in nomenclature among species, and the basis for species delimitation has not yet been standardized. Usually, the fruit body size, shape, beak shape, fruit surface verrucose projections, horn size, shape, number, and position are the main basis for the identification of extinct and extant *Trapa* [3]. So far, a total of 81 extant species and a vast number of varieties have been established [4]. However, some authors believe that there are only three extant species worldwide, or even only one polymorphic species with multiple varieties [5]. In the flora of China, Chinese *Trapa* is further divided into two species, *T**. natans* and *T**. incisa*, with the remaining species being assigned as varieties of them [6].

The fossil fruits and pollen of *Trapa* are diverse and occur throughout the Northern Hemisphere in the Cenozoic, with more than 70 species having been described to date. Seventeen species have been recorded from the Neogene of Japan [7,8,9]. About 16 species are described from Russia [10,11,12,13,14] while 16 species are described from the Miocene and Pliocene of Europe [15,16,17]. Only three species have been recorded from the Neogene of North America [11,18,19]. In China, *Trapa* fossils have been reported from the Miocene-Pliocene of Yunnan, the Late Miocene of Zhejiang, and the Miocene of Fujian [20,21].

However, no reliable *Trapa* fossil leaves have been found yet. The major reason is that the *Trapa* leaf has an inflated petiole structure, keeping its leaf blade floating on the water surface and its thin cuticles. These structures make the *Trapa* leaves more susceptible to oxidative decay and difficult for the fossilization process [3].

*Hemitrapa* Miki is an extinct aquatic genus, first established by Miki based on fruits discovered in the upper Miocene of Japan [22]. *Hemitrapa* is considered to be the only genus phylogenetically closely related to extant *Trapa* [3] and classified in the family Lythraceae together with *Trapa*. Morphologically, *Hemitrapa* is very similar to *Trapa* in fruit, both having one or two pairs of horns (arms as *Hemitrapa*) and a fruit head with parallel ridges. However, *Hemitrapa* is obviously distinguishable from the overwhelming majority of *Trapa* on the basis of its different fruit head and fruit shape, nature of its arms, and neck contraction [23]. *Hemitrapa* fossil fruits are widely distributed in the upper Paleocene to lower Pliocene of the Northern Hemisphere [8,24,25,26,27,28,29,30]. The oldest known fossil of *Hemitrapa* was found in the upper Paleocene of Canada [31]. In China, *Hemitrapa* fossil fruits were reported from the Miocene of Shandong [23], the Miocene of Fujian [21], and the Oligocene of Tibet [32].

The distribution pattern of the Paleogene climate of China is a highly debated topic. Based on palynology, Song et al. [33] initially subdivided the Paleogene climate of China into three zones, i.e., a humid warm temperate to subtropical zone in the north, a tropical to subtropical zone in the south, and an arid zone in the middle, due to the increased pollen percentages of *Ephedripites*, a representative of a xerophyte family, which is usually employed to indicate an arid climate. This was later supported by evidence from megafossil plants and lithological records [34,35]. Recently, however, Quan et al. [36] undertook quantitative Eocene climate studies based on plant fossils, which have strongly suggested a warm and humid Eocene climate in China. The mean annual precipitations over China, including areas in the so-called middle arid zone, could not have been less than 730 mm, i.e., much higher than the threshold of 500 mm, which is regarded as a precipitation boundary between semi-arid and subhumid climates [35]. In this case, the salinity content of the water bodies plays a critical role in this issue.

*Trapa* and *Hemitrapa*, as freshwater aquatic plants, require more demanding ecosystem conditions to survive, such as precipitation and evaporation, so they are eligible to qualitatively estimate the relative salinity of the water bodies that they lived in, although they are not qualified to be used to quantitively reconstruct the parameters of the paleoclimate. In this paper, we describe various fossil organs of *Trapa* (leaves, fruit, roots) from the late Eocene Bailuyuan Formation, the Weihe Basin of central China, which has the most reliable fossilized leaves of *Trapa* and its earliest fossil record. Moreover, a fruit fossil of *Hemitrapa* is also reported. The present aquatic fossils of *Trapa* and *Hemi**trapa* provide a new clue to the late Eocene ecosystem of central China.

## 2. Geological Background

The studied fossil site of the Bailuyuan Formation is located in Yangguo Town of Weinan City, Weihe Basin of Shaanxi Province, central China (34°19′32” N, 109°31′32” E; Figure 1a). The Bailuyuan Formation was established in 1959 and designated as the late Eocene–early Oligocene. The standard section is in Zhijiagou of Lantian, Shaanxi Province, about 400 m thick (Figure 1b), with grayish-white massive sandstone interspersed with brownish-red siltstone [37].

The lithology of the fossil layer consists of siltstone and mudstone. The bottom is a gray-white coarse sandstone, overlain by gray-green siltstone with brown-red thin mudstone, which belongs to the middle part of the Bailuyuan Formation. Previous lithological studies indicate that the depositional environment of the present section is lacustrine [38,39,40,41], reflecting a relatively low-energy hydrostatic environment. There are many different plant organs, including leaves, fruits, roots, seeds, and flowers, which are well preserved in this fossil flora. Furthermore, these fossil specimens are different in size, shape, and preserved orientation. Both indicate that this fossil flora is buried in situ. The plant fossil assemblage of the formation was first reported by Tao [42], in which the majority of the plant fossils are *Palibinia* Korovin, which proliferated from the late Paleocene to late Eocene or extended into the early Oligocene [43]. Combined with the mammalian fossils, such as *Sianodon bahoensis*, Palaeolaginae indet., and *Lantianius xiehuensis*, from the studied section, it suggests that the age of the middle part of the formation is most likely the late Eocene [40,44,45,46]. Moreover, according to recent advances in the stratigraphy of fossil mammals, the lower and middle parts of the Bailuyuan Formation are late Eocene while the upper part is early Oligocene (unpublished data by Prof. Zhaoqun Zhang; personal communication, 2020). Therefore, the present fossil assemblage from the middle part of the formation is the late Eocene in age.

## 3. Material and Methods

A total of 8 Lythraceae specimens were collected from the middle part of the Bailuyuan Formation near Yangguo Town, Weinan, Shaanxi province, mainly as compression and impression fossils, including one fruit, two leaves, and one root fossil of *Trapa*; one fruit of *Hemitrapa*; and some unidentifiable incomplete fruits, either *Trapa* or *Hemitrapa*.

In this paper, the morphologic descriptive terms of leaves refer to [5,47,48], and the morphologic descriptive terms of fruits mainly come from [16,20,23,29,49]. A stereo microscope (Jiangnan JSZ5B) was utilized for specimen observation and detailed identification, and a digital camera (Nikon D90) was employed for taking photos. All specimens were preserved at the Geological Experiment Testing and Analysis Center, School of Earth Science and Engineering, Xi’an Shiyou University (Xi’an, China).

## 4. Systematics

Order: Myrtales Juss. ex Bercht. et J. Presl. 1820.

Family: Lythraceae J. St. -Hil. 1805.

Genus: *Trapa* L. 1753.

Species: *Trapa natanifolia* Z. C. Han et H. Jia sp. nov.

Etymology: The specific epithet natanifolia refers to the similarity of the leaves to those of *Trapa natans*.

Holotype: WN-0-2-6A (Figure 2c).

Paratypes: WN-0-2-57 (Figure 2a), WN-0-2-6B (Figure 2b).

Repository: Geological Experimental Test and Analysis Center of Xi’an Shiyou University, Shaanxi Province, China.

Locality and stratigraphy: Bailuyuan Formation, Upper Eocene, Yangguo Town, Weinan City, Shaanxi province, central China.

Age: Late Eocene

Diagnosis: The leaf has obvious midvein. Leaf blade rhombic to deltoid, distal half of margin coarsely dentate, base broadly cuneate, petiole inflated about the middle.

Description: The leaf blade is rhombic, approximately 50 mm long and 45 mm wide. The leaf base is broadly cuneate, tip is subacute, distal half of margin coarsely dentate, proximal part is entire, net-veined, and with obvious midvein and secondary veins (Figure 2b,c,e–g). The lower part of the petiole about 20 mm from the leaf base is inflated, about 30 mm long and 6 mm wide (Figure 2a).

Genus: *Trapa* L.

Species: *Trapa* sp.

Diagnosis: The fossil consists of filiform roots.

Description: Fossil is filamentous, extending radially. The main axis is 14 mm long and has 28-33 lateral roots. The longest lateral root is 11 mm, and the shortest is 3 mm, avg. 5 mm (Figure 2d).

Genus: *Trapa* L.

Species: *Trapa* sp. indet.

Diagnosis: Fruit of medium size, with one upper horn triangular in outline, pointing upwards, the surface of the fruit head and neck finely ribbed, lower part of the body of the fruit obtriangular in outline.

Description: The fruit is obtriangular in outline, has one upper horn, length of the fruit (including neck) 12.5 mm, width at the level of the upper horn 13 mm, the upper horn 3.8 mm long, pointing upwards, with an inclination of 30°, neck protuberant, bearing neck 1.2 mm long and up to 2.5 mm broad, the surface of the fruit head and neck finely ribbed, lower part of the body of the fruit obtriangular in outline, base of the fruit gradually narrows (Figure 3a).

Genus: *Hemitrapa* Miki 1941.

Species: *Hemitrapa* sp. indet.

Diagnosis: Fruit fusiform, fruit head conical or dome-shaped, and surface finely ribbed or slightly fimbriate and constricting toward the neck along the middle of the fruit body, thin arm inserted approximately halfway along the fruit body.

Description: Fruit fusiform, 28 mm long and 17 mm wide, the fruit head is approximately 14 mm high and tapered into a conical or dome-shaped contour but does not form a distinctively contracted snout. The fruit head is finely ribbed or somewhat fimbriate on the surface, bearing one ascending thin arm on the fruit’s right. The thin arm is inserted in the proximal half of the fruit body, forming a very acute angle of 20° with the longitudinal axis of the fruit, arm up to 1.2 mm wide and 6 mm long (Figure 3d). 

## 5. Discussion

### 5.1. Morphological Comparison

#### 5.1.1. Leaf

The characteristics of the extant *Trapa* leaf blade are a rhombic to deltoid shape, leaf margin with characteristic double-mucronate tooth apices, net-veined with an obvious costa, distal half of the margin coarsely dentate, proximal part entire, leaf base broadly cuneate, with an inflated petiole, etc. [48]. The specimen WN-0-2-6A has a rhombic shape, very similar to the leaves of extant *Trapa* (Figure 2h,i) and the distal half of the margin coarsely dentate while the lower part is entire and net-veined. Extant *T. natans* (based on Flora of China taxonomy) is very similar to specimen WN-0-2-6A in leaf base characteristics while specimen WN-0-2-6B has obviously inflated petioles, such as in extant *Trapa* leaves.

Most aquatic plant leaves lack a well-developed cuticle, and the petiole of *Trapa* has an inflated structure, enabling the leaves to float on the water surface [48]. As a result of these two factors, they are more susceptible to oxidative decay. Therefore, the leaves of *Trapa* are not easily preserved as fossils [50]. Fossilized leaves of *Trapa* reported from Kazakhstan, the United States, Canada, and Poland [51,52,53,54,55,56,57,58,59,60,61] have all been transferred to *Quereuxia* Krysht [3]. In China, *Trapa* leaf fossils reported from the Cretaceous strata of Heilongjiang [62], the Eocene flora of Yilan, Heilongjiang [63], and the Miocene flora of Shanwang, Shandong [64] have also been rejected due to the lack of corresponding evidence [3]. Therefore, there is no reliable record of fossilized leaves of *Trapa* [3]. The leaf fossils found in the Bailuyuan Formation have a variety of features similar to those of extant *Trapa* leaves. Meanwhile, we adventitiously found fruit and root fossils of *Trapa* in the same stratum. We designate it as a new species, *Trapa natanifolia* Z. C. Han et H. Jia sp. nov.

#### 5.1.2. Fruit

*Trapa* fruits usually have two (to four) strongly recurved to ascending horns (indurated sepals), thick at the bases, with rough harpoon-like barbs at the tips of the horns. The variations in the size, shape, number, and position of the horns and tubercule-like projections on the fruit surface have been used to define extinct and extant species. In the fossil record, some fruits of *Trapa* and *Hemitrapa* are similar, which makes it difficult to distinguish them accurately [3,65], while each fruit of *Trap**a* is to some extent unique, although the degree of difference varies, with some specimens being lookalikes, which makes the taxonomic separation of *Trapa* species problematic [16,29,66]. The neck of our fossil fruit (specimen WN-0-2-60, Figure 3 a) is significantly contracted, with a thick upper horn base. These are the unique characteristics of the *Trapa* fruit, which is different from the fusiform fruit body of *Hemitrapa* with an inconspicuous fruit neck [23]. Therefore, it is a conclusive fossil fruit of *Trapa*. By comparison with extant and fossil species of *Trapa*, we detected that the present *Trapa* fruit has a similar fruit shape and upper horns to the extant species *T.*
*natans* (based on Flora of China taxonomy) and fossil species of *T**. srodoniana*, *T.*
*silesiaca* [67,68], *T.*
*kashmirensis* from Kashmir [69], *T.*
*kvacekii* from the Late Miocene, Greece [17], *T.*
*ninghaica**, T.*
*radiatiformis* from the Miocene, Zhejiang, China [50,70], and *T.*
*fujianensis* from the Miocene, Fujian, China [21]. Due to the incomplete preservation of the current fossil *Trapa* fruit, it cannot be further identified as any species, so we designated specimen WN-0-2-60 (Figure 3a) as *Trapa* sp. indet.

Specimen WN-0-2-26 (Figure 3d) has a fusiform fruit shape, fruit head conical or dome-shaped, and surface finely ribbed or slightly fimbriate and constricted towards the neck along the middle of the fruit body. A thin arm was inserted approximately halfway along the fruit body. The current specimen with an indistinctively contracted fruit neck and characteristically ascending arm is very similar to *Hemitrapa. shanwangensis* and *H. trapelloidea* [23,29] but unlike the *Trapa* fruit with broad upper horn bases (Figure 3a), Secondly, unlike the *Primotrapa* Y. Li et C.-S. Li found and established in the Miocene strata of Weichang, Hebei province, China [71], the arm-like structure of specimen WN-0-2-26 is attached to the middle of the fruiting body and not emanating from the base such as in *Primotrapa*. Based on these characteristics, we attributed specimen WN-0-2-26 to the extinct genus *Hemitrapa*. Since it is not well enough preserved to determine the species, we consider specimen WN-0-2-26 to be *Hemitrapa* sp. indet.

#### 5.1.3. Root

The roots of terrestrial plants usually have main roots and a well-developed root system with a thick epidermis while specimen WN-0-2-58 (Figure 2d) shows similar lateral roots and fibrous roots with only a thin epidermis. As a result, we believe that the current fossil roots belonged to an aquatic plant root system. Furthermore, by comparing the root systems with those of extant *Trapa*, it was found that extant *Trapa* has two forms of roots. One is an anchoring root, filiform and adheres to submerged mud, and the other is an assimilatory root, pinnatifid with filiform lobes [5]. Specimen WN-0-2-58 has exactly the same morphological structure as the latter assimilatory roots (Figure 2j) and is abundant in the rocks enclosing the discovered fossils of *Trapa*, so we consider specimen WN-0-2-58 as an assimilatory root fossil of *Trapa* and designate it as *Trapa* sp.

### 5.2. Systematic Affinity

The relationship between *Trapa* and *Hemitrapa* cannot be resolved through the available information. The earliest geological record of *Trapa* was Middle Miocene (Figure 4), so previous studies assumed that the modern fruit forms took shape in the middle Miocene [3]. On the other hand, the earliest records of *Hemitrapa* date from the late Paleocene in Canada (Figure 4) [31]. Since some fossils display intermediate characteristics between *Trapa* and *Hemitrapa* [13,16,21], some authors think that *Hemitrapa* may be the ancestor of *Trapa* [16,32,72,73]. Our fossil occurrences demonstrate that the *Trapa* found in the Bailuyuan Formation is the earliest in the world (Figure 4) while the *Hemitrapa* represents the earliest fossil record in Asia. Therefore, we believe modern *Trapa* originated no later than the late Eocene, rather than the Miocene, as previously thought. Our results have confirmed Miki’s hypothesis [9] that modern *Trapa* may have originated in Asia. Regarding the new occurrences of *Trapa* and *Hemitrapa*, we believe that *Trapa* and *Hemitrapa* have differentiated since at least the late Eocene.

### 5.3. Paleoecology

In previous studies, only Tao [42] reported plant fossils from the Bailuyuan Formation in the Weihe Basin of central China, most of which are *Palibinia*, and no quantitative paleoclimatic reconstructions were carried out. *Palibinia* was considered to be a typical indicator of subtropical arid climates [75,76,77,78]. However, a recent study suggests that *Palibinia* occasionally occurs in association with hygrophilous plants, such as laurels [43], which greatly reduces the validity of this taxon as an indicator of an “arid” climate [79]. In addition, the study of fossilized sporopollen in the Cenozoic of the Weihe Basin revealed that the late Eocene contained a large number of plant species that preferred warm and humid climates, such as *Rhus, Liquidambar, Podocarpus*, etc. [80]. These plants are mostly tropical subtropical deciduous broad-leaved trees or evergreen broad-leaved species, among which *Podocarpus* has a content of up to 12% [80]. This suggests that the Weihe Basin experienced a climate with an average annual temperature of 15–16 °C and precipitation of about 1500 mm, i.e., a warm and humid climate in the late Eocene [80].

Furthermore, the high content of montmorillonite in the clay mineral assemblages of the Weihe Basin compared with basins in northwestern China [81], where the clay minerals are dominated by illite and chlorite, indicates that the Weihe Basin was relatively wet during the Paleogene [81]. The strong chemical weathering and large freshwater lake deposits in the Eocene of the Weihe Basin show that it only became arid in more recent times [81]. Moreover, using the elemental boron method and the B/Ga and Sr/Ba ratios, Li [41] indicated that the Weihe Basin had a low-saline freshwater environment during the Bailuyuan period.

Extant *Trapa* mostly lives in temperate to tropical freshwater lakes and other slow-motion water bodies with water depths of 0.3–3.6 m. The optimum depth is 2 m. The temperature required for growth is about 15–30 °C, and the temperature difference between day and night should not be too extreme, as the night temperature needs to remain at least 15 °C [82]. However, currently, the Weihe Basin belongs to the warm temperate monsoon climate zone with an average annual temperature of 12–14 °C, average annual precipitation of 500–800 mm, and annual relative humidity of more than 60% [80]. Yet, there is no native extant *Trapa* in the Weihe Basin (Figure 4), which indicates that a warmer and more humid environment is required for *Trapa* to grow.

Therefore, we believe that during the late Eocene, the Weihe Basin had freshwater bodies, precipitation exceeding evaporation, and an environment warmer and more humid than modern conditions. It is possible that central China was not so arid in the late Eocene. At least, it was not persistently arid.

## 6. Conclusions

The morphological comparison reveals the world’s first definitive fossil record of *Trapa* leaves, which is designated as a new species: *Trapa*
*natanifolia* Z. C. Han et H. Jia sp. nov. The earliest fossil record of a *Trapa* fruit in the world and the earliest fossil record of a *Hemitrapa* fruit in Asia were investigated, dating back to the late Eocene. Regarding the co-occurrences of *Trapa*, as the earliest fossil record, and *Hemitrapa* in the late Eocene of central China, we believe that *Trapa* may have originated in Asia and differentiated from *Hemitrapa* at least by the late Eocene. On the basis of depositional and aquatic fossil plants’ ecological analysis, we believe that during the late Eocene, there existed freshwater ponds, lakes, and swamps in the Weihe Basin, central China, and the climate was warm and humid, which is consistent with the results of sedimentology and palynology.

## Figures and Tables

**Figure 1 biology-11-01442-f001:**
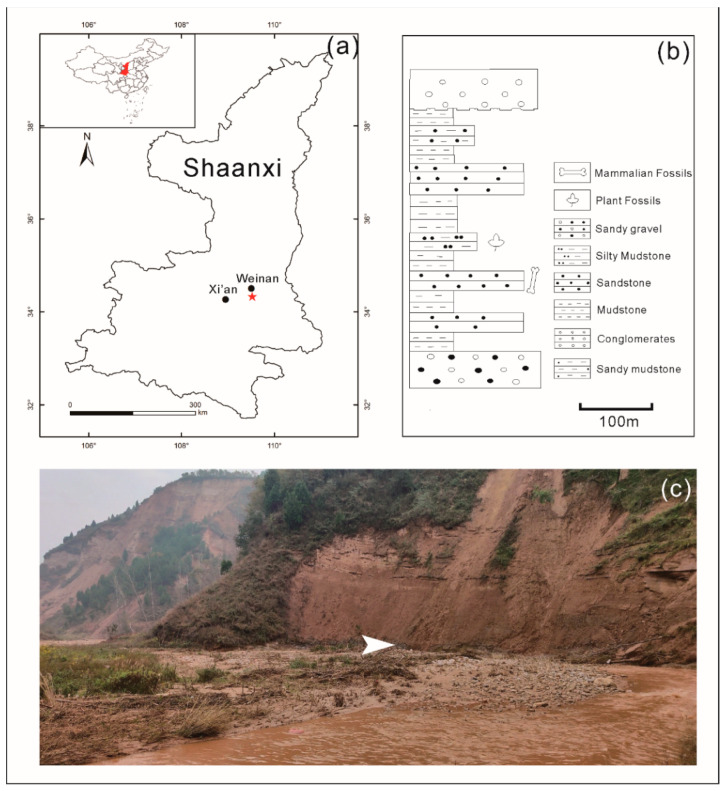
Fossil site in Yangguo Town, Weinan City, Shaanxi, central China. (**a**) The location of the fossil site. The red star represents the locality of fossil collection; (**b**) Fossil collection horizon and lithology of the Bailuyuan Formation, based on Li [37], the mammalian fossils are based on [44,45]; (**c**) The outcrop of the fossil site (white arrow).

**Figure 2 biology-11-01442-f002:**
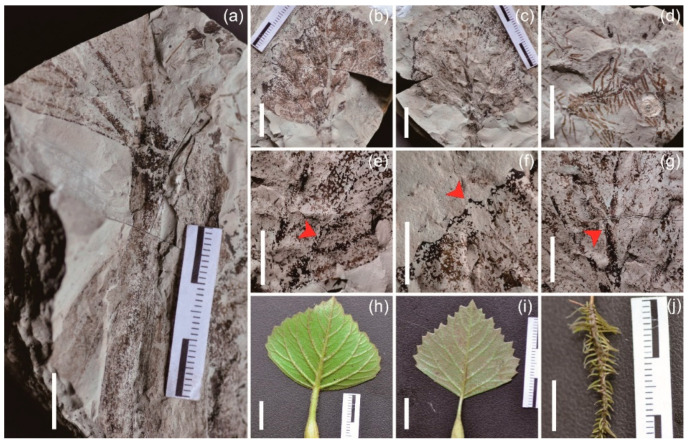
Fossil leaves of *Trapa natanifolia* Z. C. Han et H. Jia sp. nov and fossil roots of *Trapa* sp. and extant organs of *Trapa* L. (**a**) WN-0-2-57. (**b**) WN-0-2-6B. (**c**) WN-0-2-6A, holotype. (**d**) Fossil roots of *Trapa* sp. (**e**) Enlarged part of (**c**) showing secondary veins (red arrow). (**f**) Enlarged part of (**c**) showing the distal half of the margin coarsely dentate (red arrow). (**g**) Enlarged part of (**c**) showing an obvious midvein (red arrow). (**h**,**i**) Extant leaves of *Trapa* L. Scale bars = 10 mm. (**j**) extant root of *Trapa* L.

**Figure 3 biology-11-01442-f003:**
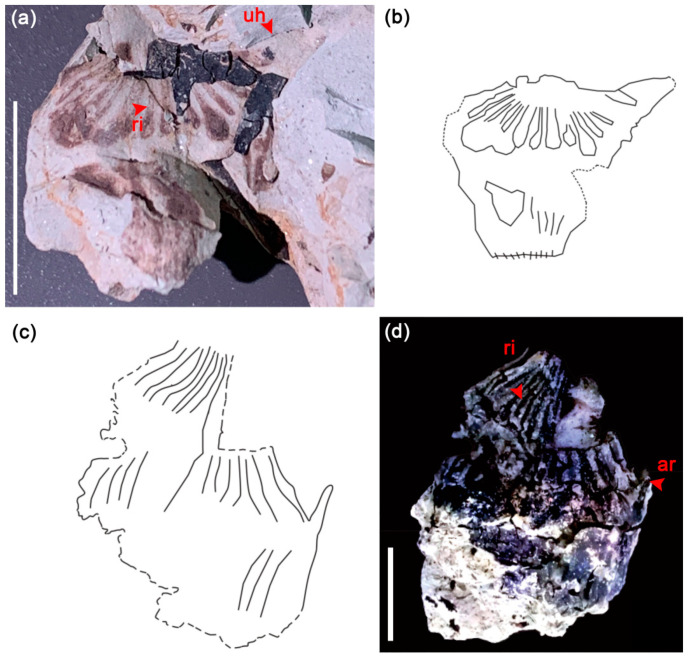
Fossil of *Trapa* sp. indet. and *Hemitrapa* sp. indet. (**a**) WN-0-2-60, fossil of *Trapa* sp. indet.; (**b**) Line drawing of *Trapa* sp. indet.; (**d**) WN-0-2-26, fossil fruit of *Hemitrapa* sp. indet.; (**c**) Line drawing of *Hemitrapa* sp. indet.; uh = upper horn, ar = arm, ri = ridges. Scale bars = 10 mm.

**Figure 4 biology-11-01442-f004:**
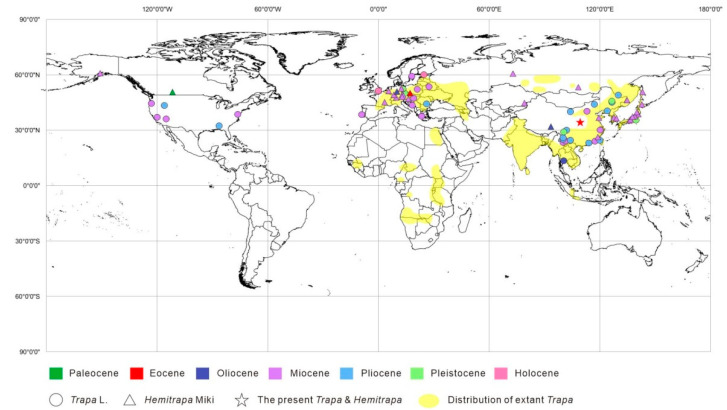
Fossil fruit records of *Trapa* L. and *Hemitrapa* Miki were reported throughout the Northern Hemisphere, and the distribution of extant *Trapa*. *Trapa* fossil records, except for China, are based on Graham [3] while those from China are based on Aung et al. [49]. The fossil distribution of *Hemitrapa* is modified from Su et al. [32]. Distribution of extant *Trapa* is modified from Mai [74]. The star represents the fossil *Trapa* and *Hemitrapa* investigated in this study. The map was created using ArcGIS 10.6.

## Data Availability

All data dealing with this study are reported in the paper.

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
