# Peer review of "A New Clue for the Late Eocene Freshwater Ecosystem of Central China Evidenced by New Fossils of *Trapa* L. and *Hemitrapa* Miki (Lythraceae)"

_biology, 2022, doi:10.3390/biology11101442_

Round 1

Reviewer 1 Report

This MS deals with very few and poorly preserved fossil specimens.

There is no information on how much material has been collected/is available to the authors.

Lines 114-123. Information on the geological frame and dating is scarce and not precise. How thick is the Bailuyuan Formation? What are its sedimentological characteristics? From which part derive the mammals? From which part derive palaeomagnetic measurements and are there any correlations to chrons? “late Eocene-early Oligocene” is quite unprecise.

Figure 1 b shows the lithology of this formation. The authors  should indicate the position of the mammal findings and palaeomagnetic results. Additionally information on the thickness should be added.

There is hardly any information about the geological/sedimentary/depositional settings and taphonomy at the fossil site and specifically of the fossil-bearing layers. Sedimentary features providing information about the depositional setting, abundance and positioning of plant remains etc. must be precise and concisely organised in a separate section/paragraph. Such information is crucial for the plausibility of ecological interpretations and conclusions.

Obviously, there is a surprising deficiency of stratigraphical knowledge of the authors!! Here are some examples:

-          Line 2 “late Eocene ecosystem”

-          Line 16 “Late Eocene-Early Oligocene Bailuyuan Formation”

-          Line 36 “Bailuyuan Formation of upper Paleogene”

-          Line 268 “Upper Proterozoic Bailuyuan Formation”

-          Lines 301-305: “Combining the above reasons, we believe that during the Late Eocene to Early Oligocene period, the Weihe Basin had quieter river bays, ox-yoke lakes or freshwater lakes, and a warmer and wetter environment. It is possible that central China was not so arid in the middle and late Paleocene, at least in terms of east-west differences in the so-called arid climate should be located west of the Weihe basin.”

…… by the way what are “ox-yoke lakes”.

Line 124 ff. “Materials and methods”: This section lacks any information on the material!!!

line 135, 136 ( and many more lines): The authors even fail to stringently apply a species-epitethon: Trapa natansfolium or T. weiheensis ?!

Line 146 ff: The “Diagnosis” lacks essential features of Trapa leaves and is not consistent with the “Description”, e.g. regarding petiole.

Line 160 ff: Is this the only root remain? It is isolated. How can the authors exclude any other systematic assignment?

Lines 160 ff, 167 ff and 182 ff.: Information about coll. numbers and quantity of material is missing.

Only two very fragmentarily preserved fruit fragments are figured. Information on further specimens is not provided. The differentiation between Trapa and Hemitrapa based solely on this material appears highly problematic.

The authors fail to put their remains into a wider context within Trapaceae. In this respect relevant are late Eocene records from Europe (reference 69) and leaf remains of Mikia (reference 41).

Lines 266 ff. Regarding the climate interpretation: Trapa is an aquatic plant and thus an azonal (or intrazonal) one. Such plants are generally of limited value for climate interpretation because their habitat (and micro climate) has stronger influence on their occrurrence than macro climate. Therefore conclusions for macro climate may be problematic.

Reference 17: This reference is quoted several times and the authors even refer to it for the descriptional terms they use. However, this website is solely in Chinese. A publication in an international journal should address a broad public. Terms for descriptions must be based on an internationally understandable medium as for leaves, e.g. Ellis, B. Daly, D.C., Hickey, L.J., Johnson, K.R., Mitchell, J.D., Wilf, P., Wing, S.L. (2009): Manual of leaf architecture. – Association of the New York Botanical Garden, 190 pp.

Figure 1: The font of the captions is too small. It is impossible to read when printed. In b the scale is missing.

Figure 2: The images of the fossil leaf material are of very poor quality. The material should be figured at least in natural size and essential details should be enlarged, all in high resolution. Otherwise, it is not possible to trace the described details.

Figure 3 The lines of the drawings of the fruit remains are too thin to appear properly when the MS is printed.

Numerous phrases are not proper phrases and it remains open what they should tell. Here are only few examples:

Lines 32-35: “It was not as arid as previously thought in central China during the Late Paleogene, either the so-called arid zone in terms of east-west division is located at least in the west of the Weihe Basin.”

Lines 100-103: “The study area is located in Yangguo County of Weinan City, Shaanxi Province, China (34 °19 °32 "N109 °31 °32" E; Figure 1 a), located in the south of Weihe Basin, belongs to the warm temperate monsoon climate zone with an average annual temperature of 12-14 .”

Lines 189-192: “At 18mm from the crown of the fruit, the angle between one arm and the main body ranges from 60° to the central axis of the fruit body, approximately 3.5 mm long, the broadest part of the fruit body where are connected with arm, about 21mm (Figure 3, d).”

Lines 203-205: “What’s more extant T. bispinosa and T. natans had much in common with specimen d in leaf base characteristics and specimen b has the air sac-like structure on the petiole of Trapa, in addition to the characteristics of modern Trapa leaves.”

The most confusing phrase probably is found in lines 267 ff: “In the Weihe Basin of central China, only Tao [9] reported plant fossils from the Upper Proterozoic Bailuyuan Formation, most of which are Palibinia, and no common Palibi-nia were previously considered to be typical of subtropical arid climates, representing subtropical arid or seasonally dry climates [10,11,78,12], however[79], concluded throug-h [80] paleoclimate simulations that the sporadic occurrences of specific arid plants may not be fully representative of arid vegetation types and may be insufficient to determine whether because they may represent dry periods throughout the humid climate[81–83] ,and there is also considerable controversy as to whether Palibinia can represent an arid climate.”

Moreover, this MS is full of spelling and linguistic mistakes. The use of upper and lower case letters and of spaces appear rather random (see e.g. the provided examples of phrases above).

There are still more issues but I stop here.

Final comments:

In case more material is available, more precise information is provided on geology, sedimentological context, age, and taphonomy and with a more profound and professional elaboration this could be an interesting contribution.

I am wondering that a native English speaking author agrees to the submission of such a poor manuscript.

Reviewer 2 Report

The data presented in the manuscript are really important for a better understanding of morphological and taxonomical aspects of fossil representatives of Lythraceae in China. However, the presentation of the text, mostly the way that the sentences were constructed in English makes the manuscript really hard to understand. So, I strongly advise the authors to carefully revise their text and make sure that a new version of it is proofread by a native speaker of the English language. I feel that most paragraphs lack clarity, meaning that the sentences should be more concise and clearer to the reader. After that, I think this manuscript could be re-evaluated concerning its scientific contents, but not as it is currently written.

Kind regards, 

Reviewer 3 Report

This study reported early fossil records of Trapa and Hemitrapa from the late Eocene to early Oligocene of central China. Since Paleogene plant fossil records in Central China is quite scarce, these new discoveries could undoubtedly increase our understanding for the plant diversity at that time. Moreover, as the early fossil records of Trapa and Hemitrapa, these fossils provide new clue for the evolutionary history of these two groups. Generally, I congratulate the authors for such valuable specimens and suggest minor revision. I think this work may be revised with my comments as below:

The title of this manuscript may be rephrased because there might be misunderstanding for ‘the earliest known organs’. Actually, Hemitrapa has quite a long fossil record worldwide dated back to the late Paleocene. Moreover, the age of these fossils is the late Eocene-early Oligocene as stated by the authors, not ‘late Eocene’ shown in the title.

 For the systematic assignment of fossil leaves, the authors need to give detailed morphological comparison between their leaf fossils and previously reported fossil records to determine the morphological evidence as a new species. For me, judging by the preservation and general shape of these two fruit fossils, these specimens may belong to the same morphotype. Only more detailed morphological observation and description are carried out could the systematic position of these fruits be better determined.

Round 2

Reviewer 1 Report

The manuscript has been improved considerably. However, there still remain some partly serious issues:

Lines 109 -110: The preservation of “a variety of plant organs” not necessarily allows the conclusion for in situ fossilisation. – Therefore this assumption requires more information.

Line 128: “About 70 specimens were collected” – what are they except the four or five described ones? I really wonder why the authors do not provide any information about the material collected. If the remains were buried in situ as the authors assume (lines 109/110) one would expect densely packed material and roots and axes in situ etc. Especially one would expect numerous remains of Trapa (especially nuts and roots) and Hemitrapa. This again rises the question why only two fruit fragments assigned to Trapa and one assigned to Hemitrapa are included in this MS. This is not at all convincing evidence for the coexistence of both genera.

Line 320: oxbow lakes ? - the Bailuyuan Formation is regarded lacustrine (see line 108).

Figure 1: In b the scale is still missing.

Figure 2: As I have noted already earlier: The leaves should be figured at least in natural size and essential details should be enlarged. Otherwise, it is not possible to trace the described details. This concerns especially the leaves figured in (b) and (d).

Minor mistakes (spaces, punctuation marks) are marked and few comments I have included in the MS.

Reviewer 2 Report

Dear authors,

thank you for improving the writing of this manuscript. It is now well-written (minor corrections still needed and highlighted in green) and presents relevant new fossils of aquatic plants of Lythraceae. I consider that after all the minor mistakes are dealt with in the text, this manuscript should be accepted for publication.

Kind regards,

Rafael Almeida
